# Into the Single Cell Multiverse: an End-to-End Dataset for Procedural Knowledge Extraction in Biomedical Texts

**Ruth Dannenfelser**[1]     **Jeffrey Zhong**[1]     **Ran Zhang**[2]     **Vicky Yao**[1]*

[1] Department of Computer Science, Rice University
[2] Department of Genome Sciences, University of Washington

## Abstract

Many of the most commonly explored natural language processing (NLP) information extraction tasks can be thought of as evaluations of declarative knowledge, or fact-based information extraction. Procedural knowledge extraction, i.e., breaking down a described process into a series of steps, has received much less attention, perhaps in part due to the lack of structured datasets that capture the knowledge extraction process from end-to-end. To address this unmet need, we present FlaMBé (Flow annotations for Multiverse Biological entities), a collection of expert-curated datasets across a series of complementary tasks that capture procedural knowledge in biomedical texts. This dataset is inspired by the observation that one ubiquitous source of procedural knowledge that is described as unstructured text is within academic papers describing their methodology. The workflows annotated in FlaMBé are from texts in the burgeoning field of single cell research, a research area that has become notorious for the number of software tools and complexity of workflows used. Additionally, FlaMBé provides, to our knowledge, the largest manually curated named entity recognition (NER) and disambiguation (NED) datasets for tissue/cell type, a fundamental biological entity that is critical for knowledge extraction in the biomedical research domain. Beyond providing a valuable dataset to enable further development of NLP models for procedural knowledge extraction, automating the process of workflow mining also has important implications for advancing reproducibility in biomedical research.

## 1 Introduction

The recent onslaught of pre-trained language models has spurred on tremendous advances in a range of natural language processing (NLP) applications, including named entity recognition (NER), named entity disambiguation (NED), sentiment analysis, and relation extraction [1–5]. These applications mostly fall under the umbrella of tasks that aim to extract *declarative knowledge*, sometimes also referred to as "knowing that," since these tasks focus on matters of factual knowledge (e.g., *knowing that* "neuron" is a cell type) [6, 7]. Declarative knowledge is often contrasted with *procedural knowledge*, or "knowing how," (e.g., *knowing how* to conduct an experiment) [6, 7]. Early AI researchers raised the importance of developing representations of procedural knowledge, given that performing plans or procedures is a fundamental way in which humans navigate the world [8]. However, compared with declarative knowledge extraction, there remains a vast gap in the development and application of machine learning methods towards procedural knowledge tasks [9].

Recently, there has begun to be a renewed interest in using machine learning to model procedural knowledge, especially knowledge extraction from text using NLP. These efforts have mostly focused

---

*Address correspondence to: vy@rice.edu

37th Conference on Neural Information Processing Systems (NeurIPS 2023) Track on Datasets and Benchmarks.

on cooking and other common household tasks [10, 11], business processes [12], and technical manuals or manufacturing [13]. The specific applications that have garnered interest seem to have been naturally motivated by either the emergence of valuable datasets (e.g., online recipes for cooking, WikiHow for various how-to tasks) or economic gain through business process optimization. Interestingly, one of the main ways scientists and engineers communicate their findings—through academic papers—is a prime source of unstructured text describing "know-how," yet few studies explore extracting procedural knowledge from scientific literature. This is the case though there is also an abundance of open access scientific literature that is frequently used for many standard declarative knowledge extraction studies.

We posit that there are 3 main reasons that procedural knowledge extraction from scientific literature is not currently widely studied:

1. Though most research papers will describe procedures, i.e., methods, they are typically not written with as much structure as a recipe or technical manual, and thus not as easy to model "off the shelf." In fact, methods sections are often organized by thematic categories and do not necessarily represent the "temporal ordering" in which the individual steps were done.[1] It is also often the case that the results sections need to be read together with the methods sections to reconstitute how various tools were used.

2. There can be varying degrees of ambiguity in a scientific manuscript when systematically describing a workflow. The same method or software tool can be used at several time points throughout a paper, but in different contexts and for different purposes. For example, principal component analysis (PCA) can be used for dimensionality reduction, feature selection, or visualization. Failure to account for context may lead to a workflow that appears to simply have a chain of PCAs. In addition, multiple parallel workflows can be described in a single paper. For example, a single paper can consider multiple datasets, each of which are processed differently, before they are analyzed jointly.

3. Unlike writing down recipes or household tasks, annotating the workflow used in a scientific paper is challenging without domain expertise, thus resulting in a bottleneck for developing structured datasets.

Motivated by these observations, we introduce FlaMBé (Flow annotations for Multiverse Biological entities). FlaMBé is a collection of structured annotations in biomedical research papers, with a particular focus on computational analysis pipelines in single cell research. While scientists have long been interested in studying single cells [14, 15], it was with the introduction of high-throughput single cell sequencing technologies around 2010 [16] that this area has exploded in activity, not only in applications of this experimental technique to various biomedical applications, but also in the development of computational tools and software to analyze the resulting data. Recent efforts to wrangle the space of analysis tools has resulted in specialized databases such as scRNA-tools [17], which currently tracks over 1,500 software tools across over 30 analysis tasks.[2] Interestingly, the majority of tools catalogued by scRNA-tools are used for more than one analysis task, and one of the most commonly used tools, Seurat [18], is associated with as many as 10 categories of tasks, further highlighting the importance of considering context.

In FlaMBé, we develop a structured representation of the procedural knowledge represented in scientific literature by considering (1) the *targets* of the study, which in the case of single cell research, are the tissues and/or cell types that are assayed; (2) the *tools* applied in the study as well as the analysis task or *context* in which they are being used; and (3) the *workflow* between tools and analysis tasks, e.g., when PCA is used for dimensionality reduction before the results are clustered using DBSCAN. Part of the motivation in structuring FlaMBé in this manner is that we can break down the more complex, unstructured goal of procedural knowledge extraction into existing, more manageable declarative knowledge extraction tasks. For example, the identification of targets and tools in text reduces to NER and NED tasks.

---

[1]Note that here, temporal ordering is used loosely, as we are simply referring to the workflow ordering that a reader can deduce from the manuscript. It is of course common that scientific manuscripts present their main results in differing order than originally conducted. That said, we expect that internal ordering of tasks within each major result to be typically a good reflection of what was actually performed.

[2]The terminology scRNA-tools uses for these analysis tasks is "categories," since they are focused on grouping tools by their applications. We simplify the terminology here to make clear that each tool can have multiple category tags.

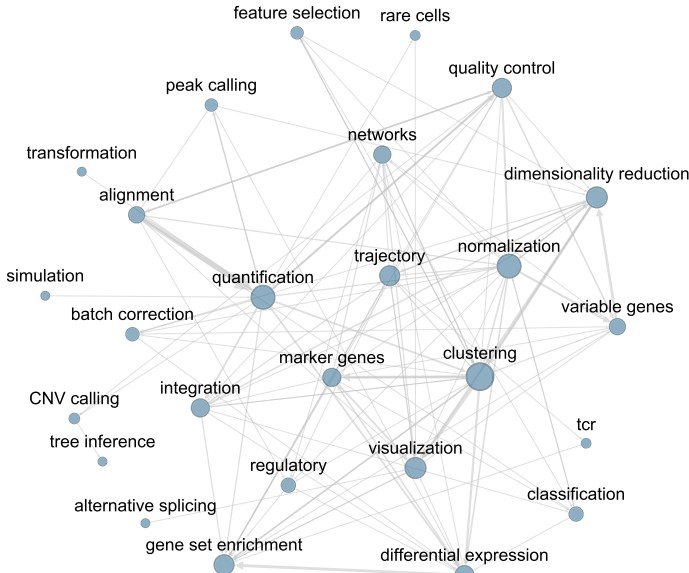

Figure 1: **Example overview of the workflow of tool contexts (analysis tasks).** Summary figure of workflows from different tool contexts captured by FlaMBé. Direction of edge represents which analysis task was completed prior to the output being sent to the following task. Weight of edges represent the number of papers that mentioned. Node size corresponds to degree, i.e., the number of papers that mentioned the corresponding analysis task.

Overall, we present 55 full text papers, including nearly 420,000 tokens, annotated for relevant entities and relations from the PubMed Central Open Access Subset by domain experts (computational biologists). To improve coverage over a more diverse set of journals and entities, we also provide tissue/cell type annotations in 1,195 paper abstracts mined from PubMed, covering over 290,000 tokens. The entire dataset provides entity annotations as well as disambiguation, where entities are linked to identifiers in relevant knowledge bases. To our knowledge, FlaMBé is the largest NER and NED dataset for tissues/cell types. Furthermore, we also provide annotations for software tools and computational methods, also capturing 28 unique contexts in which the tools are used for single cell research and nearly 400 workflow relations between (tool, context) pairs. An example visualization of the flow between contexts is shown in Fig. 1. FlaMBé is available for exploration and download at https://github.com/ylaboratory/flambe.

We illustrate some example use cases for FlaMBé here, but the richness of this dataset has many more potential downstream applications in machine learning as well as computational biology and the wider biomedical field. In general, the complexity of working with single cell data and its capacity for a variety of different workflows, together with its important biomedical applications, is ultimately what led us to choose the area of single cell research for FlaMBé. However, we have also proposed a systematic framework to distill procedural knowledge into a structured dataset in a manner that considers some of the unique challenges of scientific literature. It is our hope that FlaMBé provides a useful foundation for future "science-know-how" modeling and datasets.

## 2 Related Work

FlaMBé is designed to represent a collection of complementary tasks that together form the basis of a structured representation that captures procedural knowledge in biomedical texts. Here, we discuss related datasets and research efforts.

**Biomedical NLP**    Systematic evaluations of language models in a variety of different benchmarking efforts have revealed that for specialized domains like biomedicine, language models developed using domain-specific text (e.g., scientific literature) often outperform general-domain language models (e.g., trained on Wikipedia, news articles, webpages, etc.) on domain-specific tasks [19–22]. Furthermore, it seems that mixed-domain pretraining can sometimes hurt more than help, suggesting

that transfer learning is at times unsuccessful due to how different general-domain text is from biomedical text [20]. In general, there has been a demonstrated need for both domain-specific pretrained language models as well as domain-specific datasets for method benchmarking.

In the biomedical domain, language models are often trained on a mix of abstracts from PubMed and full text articles from PubMed Central [19–21, 23], at times also with additional scientific text such as medical records [24]. A variety of biomedical NLP benchmarking datasets have also been developed [20, 24, 25], but often individual tasks can be fragmented. Very recently, large scale efforts like BigBio [26] have systematically organized comprehensive public collections of biomedical NLP datasets. BigBio's curation revealed that the largest represented task within biomedical NLP is unsurprisingly NER, as there are a variety of biological entities that are often of interest for text mining (e.g., diseases, gene names, chemical compounds, anatomy/tissue/cell type). Of particular relevance to our work here are previous dataset curation efforts for tissue/cell type [27–30]. However, not only are these datasets smaller in terms of total annotations in comparison with FlaMBé, but furthermore, none provide NED. Disambiguating these terms and linking them to a systematic knowledge base provides more utility for the biomedical community and also enables incorporation of information from the associated knowledge base for improved knowledge extraction.

The other entity that has recently begun to be considered for NER in biomedical literature is software. As the field of computational biology grows and, accordingly, the number of software tools and computational methods, systematic identification and analysis of tool usage has become more relevant. Large-scale curation efforts for NER and NED here include bioNerDS [31], SoftCite [32], and SoMeSci [33]. These previous datasets have differing limitations. Both bioNerDS and SoftCite only consider articles published before 2011[3] in their dataset, while SoMeSci curates articles as recent as 2020. However, SoMeSci's main endpoint is a knowledge graph and thus does not provide its annotations in an easily usable format. Both SoftCite and SoMeSci have been used as training data to automatically identify software mentions across millions of scientific articles [34, 35], though the resulting automatically annotated datasets differ greatly. Finally, all previous datasets focus solely on software. Because one of the key goals of FlaMBé is to extract data processing and analysis workflows, we also wanted to expand annotations to computational methods that are often referred to in scientific papers without necessarily a specific associated software (e.g., PCA, SVM).

**Procedural knowledge extraction**   Recent efforts in procedural knowledge extraction have been spurred on by the increasing availability of naturally arising procedural knowledge-related data sources. In fact, the widespread availability of online recipes have given rise to the new research area of "food computing." [10] Other areas where there is active research in procedural knowledge extraction include household tasks based on mining data sources such as WikiHow, Instructables, and eHow [11], technical manuals [13], and business processes [12].

There has also been some limited attempts to examine scientific literature as an application area. Song et al. propose representing procedural knowledge as (target, action, method) triplets based on MEDLINE abstracts [36], and Halioui et al. consider using process-oriented case-based reasoning to extract workflows from papers mentioning phylogenetic analyses from PubMed [37]. Interestingly, these two pieces of work fall on two ends of the spectrum in terms of the complexity of the representations they propose. In addition to the limitations of modeling an entire workflow from only an abstract, Song et al.'s proposed representation is also unable to take into account when tools are applied in different contexts. Meanwhile, Halioui et al.'s representation is somewhat arduous, and their contribution is mostly focused on a rule-based workflow extraction framework rather than the assembly of a dataset that can be used by other methods. Neither Song et al. nor Halioui et al.'s datasets are accessible.[4]

## 3   Dataset Collection Methodology and Overview

Annotations for NER, NED, and other knowledge extraction tasks were curated by domain experts in computational biology for a series of 55 biomedical full text papers and 1,195 abstracts, indexed on PubMed Central (PMC) and PubMed, respectively. We chose to include both full text and abstracts

---

[3]bioNerDS further restricts its annotations to only two journals, *BMC Bioinformatics* and *Genome Biology*.

[4]Halioui et al. provide a link to their data and framework implementation in their paper, but the link is no longer active.

in FlaMBé to have a breadth of unique tokens as well as the depth needed to extract meaningful biological workflows.

## 3.1 Collection Methodology

**Abstract corpus**  The abstract corpus was hand-curated for tissue and cell type terms across 20 high-impact biomedical journals (full list in Supplementary Materials). To ensure that no single journal was overrepresented due to publication quantity, we set the number of sampled abstracts per journal to 60. Furthermore, we only sampled from recently published works between 2016 and 2021, as advances in technology have made it possible to study cell types in addition to bulk tissue and we want to capture the new diversity of cell types in our annotations. All abstracts were downloaded using PubMed eutils. To enable evaluation of interannotator agreement (Supplementary Materials), each of 3 annotators was assigned 400 abstracts (60 from each unique journal), with 240 overlapping abstracts evenly distributed across journals.

**Full text corpus**  Because of the focus on single cell research, we used Pubmed eutils to query PMC for 3 general article types ("Classical Article," "Clinical Study," and "Journal Article") using the following key words (allowing dashes to be used as a connector as well): "scRNAseq," "single cell RNAseq," "single cell RNA sequencing," "single cell transcriptomics," "single cell transcriptome." Full text articles were downloaded directly via the PMC FTP and parsed using Pubmed Parser [38]. Out of the 55 total full text articles annotated by 2 annotators, 10 papers were annotated by both to evaluate interannotator agreement (Supplementary Materials).

## 3.2 Annotation Types

Tissue, cell type, tool, and method were annotated using the Prodigy software tool developed by Explosion AI for easy tracking of token-level tags. Due to the more limited presence of tool and methods, ergo tool context and workflow in abstracts, these annotations were only completed in the full text corpus. Tissue and cell type were annotated in both the abstract and full text corpora.

**Tissue and cell type**  To determine what classifies as tissue or cell type label, we use the terms in the NCI Thesaurus,[5] a comprehensive biomedical ontology for describing human samples which has cross-references to many other biomedical ontologies, as a guide. We focus on annotating useful sample descriptors that capture what biological entity is being studied, and try to tag the most specific term possible (e.g., "left ventricle" vs. "ventricle"). The full set of annotation rules given to each annotator can be found in the Supplementary Materials.

A tissue or cell type in the text may be more specific than a term in the ontology, or it may not match exactly or any of the given synonyms. In these cases, we manually disambiguated the tag back to its nearest term in the ontology. In all other cases we programmatically mapped exact matches and synonyms back to NCIT identifiers. Additionally, in some cases, to express the specificity found in the text, we used two terms from the ontology in the disambiguation (e.g., "adipose stem cell" is mapped to two terms in NCIT "adipose" and "stem cell").

**Tool and methods**  Unlike tissues and cell types which have standardized ontologies, there is no concrete vocabulary to annotate tools and methods in biomedical research. We have done our best to define two concrete categories of methods, those where an important computational transformation of the data has taken place but can be done by more than one package, (e.g., K-means clustering or PCA), and those that reference a specific tool or package. We label each of these respective types as unspecified method ("UNS_METHOD") or tool ("TOOL"). Furthermore, we aimed to identify computational methods applied on data that are separate from sequencing technologies and their related protocols (e.g., those done on machines which physically handle a biological sample) and only annotate tools and methods starting from the initial processing off of sequencing machines.

**Tool context**  In addition to annotating tools, methods, tissue, and cell type terms in the full text we also provide a set of tool "contexts," or the analysis task that they are used for to process or augment data. This is important, as a single tool may have multiple functions or reasons that it

---

[5]https://ncithesaurus.nci.nih.gov/ncitbrowser/

was applied (Fig. 3 shows an example paper where Seurat used in 4 different contexts). For the sake of exploring the single cell multiverse, we restricted the set of modes to important functions in processing a wide variety of sequencing data. A single mention of a tool in the text can have one or more modes assigned to it based on its surrounding context. The full vocabulary for modes can be found in Supplementary Materials.

**Workflow** On a paper level, we aim to extract the various workflows done to samples, where samples are defined as an assay (e.g., scRNA-seq, ChIP-seq, BS-seq, etc) and a sample descriptor such as tissue/cell type pair. Once a unique set of samples per paper are identified we link them with tool and mode pairs from the text. Next, we annotate the flow by tabulating all edge pairs, where a pair of tools with their corresponding modes are applied to a given sample. In cases where an unspecified important transformation took place, such as an 'UNS_METHOD' we use "unspecified_mode" as a placeholder. In this way we can reconstruct and model multiple workflows in a paper when more than one sample type is used.

### 3.3 Dataset description and statistics

**Token level tags** All token level tags, such as those for tissue and cell type and tool and method annotations are released as IOB and CoNLL files. The CoNLL files contain disambiguated annotations, with the tissue and cell type tags mapped semi-manually back to NCI Thesaurus identifiers and tools disambiguated back to a standardized name. An additional description file is also provided, one for tissues and cell types, which maps NCI Thesaurus ids to names, and one for tool and method annotations, with annotations to relevant references, GitHub, or project links.

Together, the full text tag files span 55 papers and 419,949 tokens with 245 disambiguated (784 before disambiguation) tissue and cell type terms, 298 disambiguated tools (390 before disambiguation), and 48 unique general methods (134 before disambiguation). The abstract only tag files span 1,195 papers with 294,225 tokens annotated and 288 disambiguated tissue and cell type terms (662 before disambiguation).

**Tool context annotations** Mode annotations for the various tools are provided in the tool and method CoNLL files. Each mode is manually assigned using the surrounding sentence context.

**Workflow annotations** There is no predefined standard format for paper-level knowledge extraction annotations, so we split them into the following 3 files for easy parsing: A sample description and identification file, containing a listing of unique sample assay and tissue and cell type pairs; a tools applied file linking samples with the tool-mode combinations covering modes; and tool sequence file that ties pairs of tool-mode combinations together with sample identifiers. These files cover 8 unique assays, across 28 tool modes, capturing 390 tool-tool steps. There are on average, 10 workflow steps for each of the 38 papers with a defined workflow.

## 4 FlaMBé Use Cases

The diverse collection of annotations in FlaMBé enables several different use cases. We explore 3 example use cases of NER, tool context prediction, and workflow visualization before discussing other potential downstream applications.

**Use case 1: named entity recognition** We illustrate how the IOB and CoNLL files can be used to train BERT models to predict tissue and cell type mentions in biomedical abstracts. Using the full text data as training and our abstract annotations as the hold out set for evaluation, we fine-tuned some of the most popular BERT models on HuggingFace (Table 1) for NER prediction. All models perform reasonably well, with PubMedBERT [20] having the best F1 for the cell type and tissue type identification tasks. In general, the domain-specific pretrained language models do tend to perform better than the general domain models, especially when it comes to recall.

We also aim to demonstrate the utility of our annotations by comparing them with the only other easily obtainable software annotation dataset, Softcite [32], a resource that provides annotations of software mentions in full text research publications in the life sciences and economics. Here, we partition FlaMBé's full text tool annotations into two sets of full text data, holding out 11 randomly chosen

Table 1: **Predictive performance (P/R/F1 scores) of various language models.** Language models were fine-tuned on a combination of full text and abstracts and evaluated on a mixture of both text types for cell type and tissue annotations. Best performers are highlighted in bold.

| | Cell Type | | | Tissue | | |
|---|---|---|---|---|---|---|
| | Precision | Recall | F1 | Precision | Recall | F1 |
| BERT-base [39] | 0.720 | 0.848 | 0.779 | 0.775 | 0.822 | 0.798 |
| ELECTRA [40] | **0.768** | 0.837 | **0.801** | 0.815 | 0.838 | 0.826 |
| BioBERT [21] | 0.740 | 0.847 | 0.790 | 0.776 | 0.869 | 0.820 |
| BlueBERT [24] | 0.706 | 0.872 | 0.781 | 0.790 | 0.865 | 0.826 |
| BioELECTRA [23] | 0.710 | **0.875** | 0.784 | 0.803 | **0.879** | 0.840 |
| PubMedBERT [20] | 0.737 | 0.858 | 0.793 | **0.830** | 0.857 | **0.843** |

Table 2: **Predictive performance (F1 scores) of PubMedBERT on tool annotations when using Softcite or FlaMBé (excluding papers used for evaluation) as training standard.** Tool annotations from 11 full text papers were held out from FlaMBé as an evaluation standard. PubMedBERT was fine-tuned on either the entirety of Softcite annotations or the smaller FlaMBé training standard.

| | Precision | Recall | F1 |
|---|---|---|---|
| Softcite [32] | 0.415 | 0.548 | 0.472 |
| FlaMBé | **0.779** | **0.887** | **0.830** |

papers for evaluation. We use the remaining 44 papers from FlaMBé and the entirety of Softcite for training. Both datasets were used to train PubMedBERT, one of the consistent performers in tissue/cell type prediction (Table 2). Despite being a smaller set of annotations, FlaMBé outperforms Softcite, especially when it comes to identifying the full name of a tool, (e.g., "Search Tool for the Retrieval of Interacting Genes/Proteins," more commonly known as "STRING"). This observation seems to be supported when we examine the predictive performance broken down by tag type—the largest performance difference between a model trained on Softcite and FlaMBé is in the 'I-Tool' token (see Supplement). We hypothesize that the fact that biomedical tools often have long, multi-word names (and corresponding acronym) may play in role in this large difference. Of course, we note that in this comparison FlaMBé has the advantage of using the same annotation criteria in both the training and test sets, but nevertheless, we believe it still illustrates the importance and utility of FlaMBé's biomedical specific tool annotations.

**Use case 2: tool context prediction**    As a proof of concept, we also used FlaMBé's tool context annotations and trained a PubMedBERT model to predict a tool's context given the sentence in which it is mentioned, akin to sentiment classification. We assembled a small set of training (191 sentences over 28 papers) and test (45 sentences over 8 papers) data, limiting ourselves to sentences containing a mention of at least one of the top 5 most mentioned tools, *Seurat, Cellranger, t-SNE, Monocle*, and *STAR*, each of which can be applied in multiple contexts. We then trained PubMedBERT models to predict context for each sentences in a one vs rest framework, for contexts that are well represented in the test and training datasets: *Alignment, Marker Genes, and Clustering*. Each of the classifers performed well, with the alignment (AUC = 0.954) and marker gene (AUC = 0.953) contexts being more distinguishable and clustering (AUC = 0.810) being the most difficult. Given this promising performance on a test case, we anticipate that more sophisticated methods will be able to achieve consistently strong performance with our annotations.

**Use case 3: visualization and exploration of different scientific workflows**    Different workflows can be extracted from FlaMBé's annotations, at different levels of specificity, either by highlighting the different tools used in a paper (Fig. 2A) or the different tool contexts in a paper (Fig. 2B). These can also be combined to extract more exact methodology (Fig. 3). Benchmarking papers or work introducing a new tool have to compare with previous work and create interesting workflows, as a small set of sample types is processed with slight variations through different levels of an entire pipeline depending on a paper's objective (Fig. 2). Meanwhile, papers that seek to solve a biological problem often have a more defined flow, with fewer tools from sample to one or more endpoints (Fig.

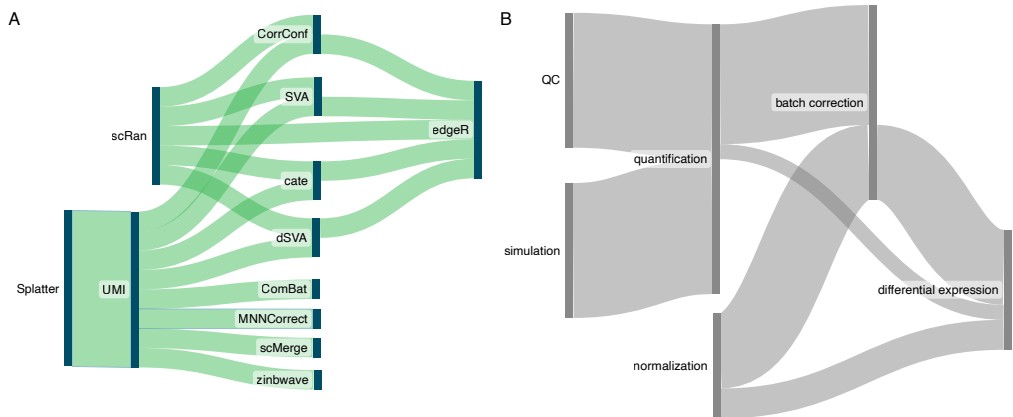

Figure 2: **Sankey visualizations of (A) tool and (B) context workflows from an example paper [41].** Visualizations here focus on one entity at a time, either the computational tools being used throughout the paper (vertical bars in A) or the context (vertical bars in B) in which they are being used.

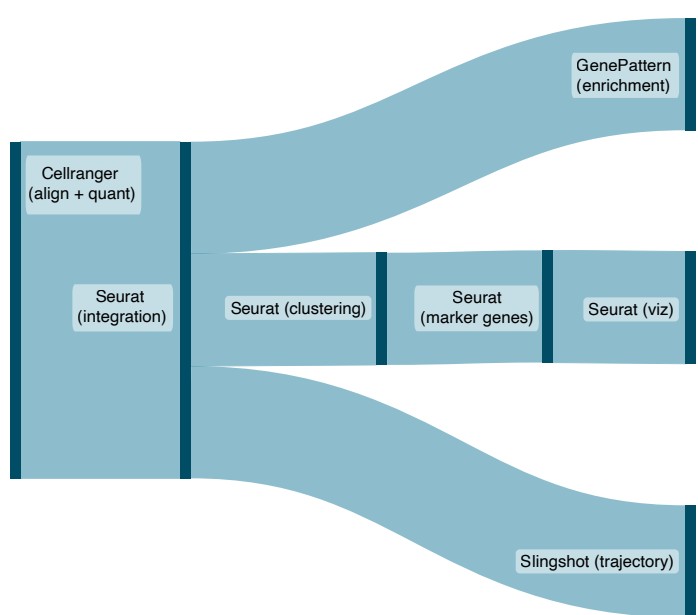

Figure 3: **Sankey visualizations of the joint tool-context workflow from an example paper [42].** Visualization here depicts the workflow of (tool, context) pairs (vertical bars), where context is denoted within the parentheses.

3). By extracting these workflows, we can not only classify the type of paper (e.g., benchmarking, new method, or biological insight), and analyze them on an individual level, but can also look at the global set of workflows for a large set of papers (Fig. 1). Thus, FlaMBé has important downstream potential for extracting knowledge at multiple levels.

## 4.1 Potential downstream applications

There are many other interesting downstream applications that FlaMBé can be used to study. In addition to the advances in developing systematic methods for procedural knowledge extraction, we want to highlight the scientific value of improved modeling here. Specifically, structured representations would potentially allow for improved computational method recommendation depending on the goals of a particular study, as well as highlight gaps and areas of need for new computational method development. Importantly, one of the natural concerns that has been raised in psychology and more

recently in machine learning is that having ever more complex computational workflows can spawn *multiverses*. The multiverse represents the set of parallel universes where slightly different paths (e.g., methods or analysis steps) are taken towards the same goal. Multiverse analyses are undertaken to see how reliable results and conclusions are in light of these implicit decisions [43, 44]. We believe one of the most exciting downstream applications of FlaMBé is systematic multiverse analyses of the complex workflows undertaken in biomedical research, towards the ultimate goal of improving transparency and reproducibility of research claims.

## 5    Limitations and Future Work

One of the current limitations of FlaMBé is that though the number of entity-level annotations is high, there are relatively fewer examples of the more complex annotation types of tool context and workflow. We plan to address this through larger annotation efforts that will further expand these categories. Because FlaMBé has also proposed a systematic, structured representation that can be used as input to existing language models, these future efforts can be aided by computational predictions that can guide manual curation efforts. In these follow-up efforts, we foresee that the NER-related annotations will be easiest to automate, followed be NED, with the tool context and workflow predictions being more challenging. Any automated annotations will be reviewed by expert curators before release of an updated dataset. We do not foresee negative societal impacts, though incorrect workflows could potentially be misleading for downstream research, and thus we would encourage thorough evaluation of all predictions.

With FlaMBé, we have broken down the more complex, abstract procedural knowledge extraction problem into more structured declarative knowledge tasks that the community is already well-equipped to tackle. Intriguingly, cognitive psychology research has pointed towards the fact that in humans, procedural and declarative knowledge are intertwined, but can sometimes be learned independently of one another [45]. Thus, there may also be benefit to using different, more "procedural" representations for learning. In some sense, one ML area that has tried to learn and mimic human procedural knowledge is reinforcement learning. A good example of this is with "script knowledge" [46] and generally text-based games [47], which have used a game approach to improve modeling at the intersection of language understanding and complex decision-making. Reinforcement learning has also found some early success in reasoning over large scale knowledge graphs. Procedural knowledge extraction from academic texts could potentially also benefit from this type of framework. One of the unique aspects of FlaMBé is that though we have developed a structured representation, they can also tie together (e.g., we have annotated individual edges that can be viewed jointly as a graph). The disambiguated terms also tie in with existing knowledge bases that can be incorporated into knowledge graph research. It will be interesting to see whether new methods can be developed that could take advantage of the joint representation and learn more than the sum of the parts.

## 6    Conclusion

In conclusion, we have developed FlaMBé, a collection of datasets that together form structured representations of procedural knowledge captured in scientific literature. The dataset provides annotations for 1,195 paper abstracts and 55 full text papers, spanning over 700,000 tokens. In addition to providing the largest NER and NED dataset for tissue and cell type, we also provide annotations for computational tool and method, as well as the analysis task a tool is used in. Finally, we also annotate computational workflows within papers that can potentially be used in many downstream applications. Our dataset and associated code are accessible at https://github.com/ylaboratory/flambe.

## Acknowledgements

This work was supported by the Cancer Prevention & Research Institute of Texas (CPRIT RR190065). VY is a CPRIT Scholar in Cancer Research.

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
