# Into the Single Cell Multiverse:
# an End-to-End Dataset for Procedural Knowledge Extraction in Biomedical Texts

**Ruth Dannenfelser**[1]     **Jeffrey Zhong**[1]     **Ran Zhang**[2]     **Vicky Yao**[1]*

[1] Department of Computer Science, Rice University
[2] Department of Genome Sciences, University of Washington

## A  Dataset and code distribution

### A.1  Link to the dataset

All annotations, disambiguation files, and corresponding code are freely accessible on GitHub: `https://github.com/ylaboratory/flambe`. A permanent archive of all data is hosted on Zenodo (linked to from GitHub).

### A.2  License

All code and annotations are distributed under CC BY 4.0. However, source text from PubMed and PubMedCentral retain their original licenses. License information was obtained from PubMed and PubMedCentral and can be found on our GitHub repository. Please note that there are some abstracts from PubMed that did not provide license information and likely retain their original copyright.[1]

## B  Annotation Methodology

Here we describe the additional details of FlaMBé's curation including structured guidelines for each annotation task, corpus curation, and file assembly.

All manual curation in FlaMBé was conducted by three annotators who have doctorate level expertise in computational biology. For named entity tagging annotations a set of structured guidelines were followed to ensure consistency. The guidelines given to reviewers are in the annotator guidelines section below.

### B.1  Tissue and cell type entities

Generally, all terms, related synonyms, and text entities that can be mapped to an entry from the *tissue, organ, body part, fluid*, and *cell type* branches of the NCI thesaurus were labeled. Instead of a rigid vocabulary fixed on exact matches of NCI Thesaurus (NCIT) terms and synonyms, annotators were encouraged to tag any word with the same meaning as an ontology term. For example, "Pancreatic ductal adenocarcinoma" describes cancer of the pancreas, which can be related back to the NCI Thesaurus, and thus was tagged as a "TISSUE". An initial set of rules was provided to each annotator. When one annotator encountered a corner case (e.g., "is neuron a tissue or cell type?") all annotators discussed, reached a consensus, then added the corner case to the set of annotation rules. The final set of rules are as follows:

---

*Address correspondence to: vy@rice.edu

[1] `https://www.nlm.nih.gov/databases/download.html`

37th Conference on Neural Information Processing Systems (NeurIPS 2023) Track on Datasets and Benchmarks.

### B.1.1 Annotation guidelines

General annotation rules

- abstracts without any entities are informative and should be included in the corpus

- to maintain annotation quality, do not index text where the tissue / cell types terms are too ambiguous in their mapping to the NCI thesaurus.

- try to capture the most specific label possible (e.g., left ventricle vs ventricle, and right side of the heart versus heart)

- tag terms that may be in an non-stemmed form (e.g., left ventricular should still be tagged as tissue)

- do not label agencies or journal names as tissues or cell types (e.g., American Heart Association, the journal Neuron)

Acronyms

- tissue acronyms should be tagged for abbreviations that refer back to any tissue entity, likewise for acronym that can be resolved to a cell type

- the acronym must only be for the tissue or cell type with no extra words

Tissue

- vague anatomy qualifiers that refer to one or more tissue types (e.g., cervical can refer to neck or the uterine cervix) should not be annotated

- unambiguous anatomy qualifiers (e.g., colorectal (in colorectal cancer)) should be tagged as a tissue

- blood treatments should not be tagged as blood, but something like blood draws if blood is the major topic in the paper or blood pressure should be annotated

- tissue terms that are a side effect should still be labeled

- references to anatomical terms when they occur as verbs or as irrelevant sample descriptors such as "arm" in "treatment arm" should not be tagged

- axon is a tissue

- cortex when alone (and given the appropriate context) refers to cerebral cortex and should be tagged as a tissue

- mouth should not be considered oral cavity but rather vestibule of the mouth which falls under body part and should be tagged

Cell type

- neurons should be labeled as cell types

- lymphocytic should not be labeled as it might not be used in tissue / cell type specific way

### B.1.2 Abstract corpus

To assemble a corpus of diverse abstracts we queried NCBI eutils for a subset of 20 journals with an impact factor higher than 10: *Annals of Internal Medicine, Cancer Cell, Cell, Clinical Oncology, Genome Biology, Immunity, Journal of the American Medical Association, The Lancet, Molecular Systems Biology, Nature, Nature Biotechnology, Nature Communications, Nature Genetics, Nature Medicine, Nature Methods, Nature Neuroscience, Neuron, New England Journal of Medicine, Science*, and *Science Translational Medicine*. This journal subset was chosen to ensure high quality works that cover a range of biomedical topics and linguistic styles, from medicine, computational biology, and experimental bench science.

## B.2 Tool and unspecified method entities

Unlike tissue and cell type entities, biomedical methods and software packages do not have a well defined structured ontology to form a consistent vocabulary. As a result, we loosely defined two categories of methods, ones that we deem "tools", which are text entries that can be linked to a single paper describing a novel computational or statistical method, and "unspecified method", ("uns_method" for short) which describe an important part of a computational workflow but may be apart of multiple methods or are implemented in more than one library. Base functionality, such as statistical tests and functions that are included in the base environment and packages for R, Python, etc are not considered as unspecified methods. To that end, reviewers also followed these additional guidelines:

- PCR, ChIP-seq, SMART-seq, or other sequencing technologies or sequencing libraries are not tools (since the focus here are software tools)
- default commercial software on sequencing machines or used by core facilities should be excluded from annotations
- the full name of a tool as well as its acronym should be tagged as tools
- count-based transformations such as UMI does not qualify as a tool or unspecified method
- do not annotate language / general framework (e.g., R, python, bioconductor, conda, tensor-flow, keras, etc)

## B.3 Workflow annotations

Annotators were given a set of pre-tagged full-text papers with tissue, cell type, and tool entities. They were then tasked with generating three files:

- "sample" file linking any experimental assay (e.g., RNA-seq, single cell RNA-seq, ChIP-seq) with tissue and cell type annotations
- "tools_applied" file joining samples, tools, and the tool context
- "sequence" file that captures pairs of applied tools

Each of the three files start each new line with PMC identifiers linking defined annotations with relevant papers. Furthermore, the "sample" and "tools_applied" files have sequential id numbers within each PMC for the extraction of unambiguous sample workflows.

All fields in the workflow annotations were constrained to predefined vocabularies. In the case of samples, tissues and cell types are linked to entries in NCI thesaurus through disambiguation, and the small set of extracted assays are standardized between all reviewers. Tools are standardized as described in tool disambiguation. When unspecified_methods are applied, a special "unspecified" tag is appended before writing the method name (e.g., "unspecified_pca"). The context in which a tool is applied is defined from a subset of categories derived from the scRNA-tools database [9]. Here we restrict the vocabulary to: *Alignment, Alternative Splicing, Batch Correction, Classification, CNV calling, Clustering, Deconvolution, Differential Expression, Dimensionality Reduction, Gene Enrichment / Gene set analysis, Integration, Imputation, Marker Genes / Feature Selection, Networks, Normalization, Quality Control, Quantification, Rare Cell Identification, Simulation, TCR, Tree Inference, Visualization, Variable Genes.*

Workflow annotations are more subjective than spans which have clearly defined boundaries. Thus, to ensure the consistency and quality of these annotations, all curated workflow annotations required agreement amongst all annotators and were established via round table discussion. Any conflicting annotations were kept only when a mutual agreement could be reached for their inclusion or exclusion.

## C Interannotator Agreement

To assess consistency across annotators we calculated Fleiss-Kappa before resolving conflicts and merging annotations into a final set. In general, annotators were consistent across the abstract ($\kappa =$ 0.807) and full-text ($\kappa = 0.875$) tissue and cell type tagging tasks, with the most common discrepancies being small differences in term specificity (e.g., "peripheral blood" vs "blood") or mixing up tissue vs

cell type labels. For tool annotations, the annotators were quite consistent labeling tools ($\kappa = 0.891$), but a little less so for unspecified methods ($\kappa = 0.769$). Since flow annotations were combined via consensus we did not attempt to calculate interannotator agreement for these annotations.

## D    FlaMBé Use Cases: Supplementary Materials

### D.1    Named entity recognition

All experiments were run on an NVIDIA A100 40GB GPU. Each epoch typically would take ∼2-5 minutes to run.

### D.2    Tool context prediction

All experiments were run on an NVIDIA A100 40GB GPU. Each epoch typically would take ∼2-5 minutes to run.

## E    Additional experimental results

### E.1    Tag-level performance metrics for the tissue and cell type labeling task

We report tag-levele performance metrics in the following tables.

Table 1: **Tag-level precision of various language models.** Language models were fine-tuned on a combination of full text and abstracts and evaluated on a mixture of both text types for cell type and tissue annotations. Per-tag performance is shown, where the 'B-' prefix represents a token that is at the beginning of a chunk corresponding to the entity, and the 'I-' prefix represents a token that is inside the chunk. Best performers are highlighted in bold.

|  | B-Cell Type | I-Cell Type | B-Tissue | I-Tissue | O |
|---|---|---|---|---|---|
| BERT-base [2] | 0.777 | 0.821 | 0.850 | 0.550 | 0.998 |
| ELECTRA [1] | **0.794** | **0.875** | 0.863 | 0.661 | 0.997 |
| BioBERT [7] | 0.791 | 0.862 | 0.827 | 0.604 | 0.998 |
| BlueBERT [8] | 0.745 | 0.806 | 0.843 | 0.600 | **0.999** |
| BioELECTRA [6] | 0.745 | 0.848 | 0.833 | **0.733** | **0.999** |
| PubMedBERT [5] | 0.783 | 0.843 | **0.888** | 0.659 | 0.998 |

Table 2: **Tag-level recall of various language models.** Language models were fine-tuned on a combination of full text and abstracts and evaluated on a mixture of both text types for cell type and tissue annotations. Per-tag performance is shown, where the 'B-' prefix represents a token that is at the beginning of a chunk corresponding to the entity, and the 'I-' prefix represents a token that is inside the chunk. Best performers are highlighted in bold.

|  | B-Cell Type | I-Cell Type | B-Tissue | I-Tissue | O |
|---|---|---|---|---|---|
| BERT-base [2] | 0.879 | 0.904 | 0.871 | 0.756 | 0.995 |
| ELECTRA [1] | 0.860 | 0.877 | 0.875 | 0.744 | **0.996** |
| BioBERT [7] | 0.888 | 0.867 | **0.914** | **0.925** | **0.996** |
| BlueBERT [8] | 0.897 | **0.936** | 0.898 | 0.919 | 0.995 |
| BioELECTRA [6] | **0.908** | 0.896 | 0.909 | 0.825 | 0.995 |
| PubMedBERT [5] | 0.881 | 0.905 | 0.897 | 0.919 | **0.996** |

Table 3: **Tag-level F1 of various language models.** Language models were fine-tuned on a combination of full text and abstracts and evaluated on a mixture of both text types for cell type and tissue annotations. Per-tag performance is shown, where the 'B-' prefix represents a token that is at the beginning of a chunk corresponding to the entity, and the 'I-' prefix represents a token that is inside the chunk. Best performers are highlighted in bold.

| | B-Cell Type | I-Cell Type | B-Tissue | I-Tissue | O |
|---|---|---|---|---|---|
| BERT-base [2] | 0.825 | 0.869 | 0.860 | 0.637 | 0.997 |
| ELECTRA [1] | 0.826 | **0.876** | 0.869 | 0.700 | 0.997 |
| BioBERT [7] | **0.836** | 0.864 | 0.868 | 0.731 | 0.997 |
| BlueBERT [8] | 0.814 | 0.866 | 0.870 | 0.726 | 0.997 |
| BioELECTRA [6] | 0.819 | 0.872 | 0.869 | **0.776** | 0.997 |
| PubMedBERT [5] | 0.829 | 0.873 | **0.893** | 0.768 | 0.997 |

### E.2 Additional performance metrics for the tool prediction task

In Table 4 we show tag level performance to highlight how multi-word tool names are captured.

Table 4: **Predictive performance (F1 scores) of PubMedBERT on tool annotations when using Softcite or FlaMBé (excluding papers used for evaluation) as training standard.** Tool annotations from 11 full text papers were held out from FlaMBé as an evaluation standard. PubMedBERT was fine-tuned on either the entirety of Softcite annotations or the smaller FlaMBé training standard.

| | B-Tool | I-Tool |
|---|---|---|
| Softcite [3] | 0.514 | 0.314 |
| FlaMBé | **0.833** | **0.752** |

## F    Datasheet for FlaMBé

Datasheet items are based off of suggested guidelines in [4].

### F.1    Motivation

**For what purpose was the dataset created?**  To explore procedural knowledge extraction in biomedical literature and to enable analysis of computational workflows in single cell research.

**Who created the dataset and on behalf of which entity?**  The authors listed on this paper, which includes researchers from Rice University and the University of Washington.

**Who funded the creation of the dataset?**  FlaMBé was funded by the Cancer Prevention and Research Institute of Texas (CPRIT RR190065).

### F.2    Composition

**What do the instances that comprise the dataset represent?**  Annotations are mostly at the token level (roughly corresponding to individual words in the abstracts and full text articles).

**How many instances are there in total?**  As discussed in the main text, the PubMedCentral full text articles include 419,949 tokens spanning 55 papers, while the PubMed abstracts include 294,225 tokens spanning 1,195 papers.

**Does the dataset contain all possible instances or is it a sample (not necessarily random) of instances from a larger set?**  FlaMBé includes a subset of PubMed abstracts and PubMedCentral full text articles.

**What data does each instance consist of? Is there a label or target associated with each instance?**
For each abstract instance, we provide annotations for tissue/cell type and disambiguation to entries in the NCI Thesaurus[2] in a `conll` file. For each full text instance, we provide annotations and disambiguation for tissue/cell type as with the abstracts, but also software tools and computational methods as well. Software tools are disambiguated to packages whenever possible, and computational methods are disambiguated to Wikipedia entries whenever possible. The full text annotations are also provided as a `conll` file. In addition, FlaMBé also includes tool context and workflow annotations for a further subset of the full text papers included.

**Is any information missing from individual instances?** Abstracts only have tissue/cell tyep annotations, so all other annotations are missing. For the full text articles, no entity annotations are missing, but tool context and workflow annotations are only provided for a subset of full text articles.

**Are relationships between individual instances made explicit?** Yes. Using the `conll` file format, the article ID is also included, and beyond that, textual context is also clear.

**Are there recommended data splits?** We used simple random stratified sampling (stratified by article) and recommend doing so for future studies as well.

**Are there any errors, sources of noise, or redundancies in the dataset?** The authors have thoroughly checked the annotations provided as a part of FlaMBé, but there is a possibility there may still be some human annotation error in the dataset.

**Is the dataset self-contained, or does it link to or otherwise rely on external resources?** The entity annotations are self-contained (as the raw text is provided in the `conll` files), but the disambiguation links to external resources (NCI Thesaurus, software repositories, Wikipedia).

**Does the dataset contain data that might be considered confidential?** No.

**Does the dataset contain data that, if viewed directly, might be offensive, insulting, threatening, or might otherwise cause anxiety?** No.

## F.3 Collection Process

**How was the data associated with each instance acquired? What mechanisms or procedures were used to collect the data? If the dataset is a sample from a larger set, what was the sampling strategy?** These are all discussed in detail in Appendix B.

**Who was involved in the data collection process (e.g., students, crowdworkers, contractors) and how were they compensated (e.g., how much were crowdworkers paid)?** The authors in the author list were involved in the data collection process; compensation is in the form of research credit.

**Over what timeframe was the data collected?** The annotations were collected at various time-points between September 2021 and June 2023.

**Were any ethical review processes conducted?** N/A

## F.4 Preprocessing/cleaning/labeling

**Was any preprocessing/cleaning/labeling of the data done (e.g., discretization or bucketing, tokenization, part-of-speech tagging, SIFT feature extraction, removal of instances, processing of missing values)?** Prodigy[3] was used for the labeling of the data, and the default spaCy English tokenizer was used for tokenization.

---

[2] https://ncithesaurus.nci.nih.gov/ncitbrowser/
[3] https://prodi.gy

**Was the "raw" data saved in addition to the preprocessed/cleaned/labeled data?** Yes.

**Is the software used to preprocess/clean/label the instances available?** Yes. Prodigy is still available and well-maintained (researchers at universities can apply for free licenses for research purposes).

## F.5 Uses

**Has the dataset been used for any tasks already?** We provide some example use cases in the main text of this manuscript.

**Is there a repository that links to any or all papers or systems that use the dataset?** N/A

**What (other) tasks could the dataset be used for?** We discuss some other downstream applications of FlaMBé in the main text as well.

**Is there anything about the composition of the dataset or the way it was collected and preprocessed/cleaned/labeled that might impact future uses?** No.

**Are there tasks for which the dataset should not be used?** No.

## F.6 Distribution

**Will the dataset be distributed to third parties outside of the entity (e.g., company, institution, organization) on behalf of which the dataset was created?** The dataset will be publicly available.

**How will the dataset will be distributed?** The dataset is available as discussed above (Appendix A.1).

**When will the dataset be distributed?** The dataset will be distributed by September 2023.

**Will the dataset be distributed under a copyright or other intellectual property (IP) license, and/or under applicable terms of use (ToU)?** Please see Appendix A.2 for more details.

**Have any third parties imposed IP-based or other restrictions on the data associated with the instances?** As discussed in Appendix A.2, some of the text (especially the PubMed abstracts) have differing publisher licenses/copyright and should be treated accordingly. PubMedCentral full text are all open access, but some are restricted for commerical use. All license information for the dataset can be found on the project GitHub repository.

**Do any export controls or other regulatory restrictions apply to the dataset or to individual instances?** N/A

## F.7 Maintenance

**Who will be supporting/hosting/maintaining the dataset?** The authors of the paper.

**How can the owner/curator/manager of the dataset be contacted?** The corresponding author can be contacted at `vy at rice.edu`.

**Will the dataset be updated?** We plan to update this dataset at least one more time within the next 2 years. Updates will be released on GitHub.

**If the dataset relates to people, are there applicable limits on the retention of the data associated with the instances?** N/A

**Will older versions of the dataset continue to be supported/hosted/maintained?** Yes. We will track previous releases both on GitHub and archived on Zenodo.

**If others want to extend/augment/build on/contribute to the dataset, is there a mechanism for them to do so?** Yes. Our license (CC BY 4.0) permits others to extend/augment/build on/contribute to the dataset.