# OpenReview forum: "Into the Single Cell Multiverse: an End-to-End Dataset for Procedural Knowledge Extraction in Biomedical Texts"
_NeurIPS.cc/2023/Track/Datasets_and_Benchmarks — NeurIPS 2023 Datasets and Benchmarks Spotlight_

### Official Review · Reviewer_FQ7K · 2023-07-20
**FlaMBé is a well designed, unique, medium-impact dataset for NER, NED and process annotation derived from scientific literature on single cell analysis.**

**Rating:** 8
**Confidence:** 4

**Strengths:**

FlaMBé provides a unique and useful resource for a number of NLP tasks including NER, NED, and context prediction (the latter being specific to procedural knowledge). The authors demonstrate that training PubMedBERT on FlaMBé records for tool annotations improves performance compared to PubMedBERT trained on Softcite records (Table 2). They also demonstrate using FlaMBé data to train and evaluate BERT models for NER tasks. The combination of those tasks with visualizations of scientific workflows to categorize study types is also an interesting use case.

The quality of the research is high. There are limited ethical and social implications of the work.

**Additional Feedback:**

There is a typo on page 6: "the targets of the study, which in the case single cell research, are the tissues and/or cell types that are assayed..." should be "the targets of the study, which in the case OF single cell research, are the tissues and/or cell types that are assayed..."

**Clarity:**

The paper is very well written.


**Correctness:**

Yes, the claims in the submission are correct. The dataset was constructed in a sound way, described in the Methods and with the annotator guidelines and inter-annotator agreement provided as supplementary materials.


**Documentation:**

Yes, there is sufficient detail provided on data collection and organization, including documentation in a dedicated GitHub repository. The dataset creation process is described in the Methods. Annotator guidelines and inter-annotator agreement provided are as supplementary materials. The authors state that a permanent version of the dataset will be hosted on Zenodo, upon acceptance.

**Ethics:**

I have no ethical concerns with the submission.


**Limitations:**

The work would be strengthened by a proposal for how to maintain an up to date instance of FlaMBé -- will it always rely on 100% manual annotation? The manuscript states that they will undertake "larger annotation efforts" but no detail is provided.


**Opportunities For Improvement:**

The authors point out that FlaMBé contains relatively few examples of complex annotations of tool context and workflow. The relevance to the broader research community is only moderate -- the dataset is relatively small (55 full text papers, just over 1000 abstracts) and static.


**Relation To Prior Work:**

Yes, the authors summarize related contributions in Biomedical NLP, including BigBio, datasets related to cell and tissue types, and datasets that capture details about software tools used in single cell research. They also describe related work in procedural knowledge extraction in general e.g. 'food computing', and applied to scientific literature specifically. FlaMBé differs in terms of the context it provides for capture software usage as part of experimental methods, and by being open access.


**Summary And Contributions:**

The authors present FlaMBé, a human curated dataset of procedural knowledge about experimental methods derived from scientific literature describing single cell research including sequencing and transcriptomics. FlaMBé captures details about tissue and cell types as well as analysis tools and the context in which they are used.

FlaMBé is derived from 55 full text papers and 1,195 abstracts, and to the authors' knowledge provides the largest NER and NED data for cell and tissue types.

---

> ### Author Response · Authors · 2023-08-23
>
> Thank you for this review!
>
> > The work would be strengthened by a proposal for how to maintain an up to date instance of FlaMBé -- will it always rely on 100% manual annotation? The manuscript states that they will undertake "larger annotation efforts" but no detail is provided.
>
> We have now added more details in the edited manuscript describing more details about follow-up efforts that would incorporate ML-guided predictions together with manual checking by expert curators (lines 314-317).
>
> > There is a typo on page 6: "the targets of the study, which in the case single cell research, are the tissues and/or cell types that are assayed..." should be "the targets of the study, which in the case OF single cell research, are the tissues and/or cell types that are assayed..."
>
> Thanks for catching this! We have fixed the typo on page 6.

---

### Official Review · Reviewer_oXH1 · 2023-07-20
**A novel dataset for extraction of procedural knoweldge in biomedical papers**

**Rating:** 8
**Confidence:** 3
**Clarity:** The paper is well-written and I found…

**Strengths:**

- I think an annotated corpus in BioNLP for procedural knowledge is great. Anecdotally I find that biologists/computational biologists are sometimes underwhelmed with current BioNLP, partly because it ignores much of the context around an experiment, biomedical entity or relation. A labelled dataset for modelling procedural knowledge should help the community build new NLP tools to address this shortcoming.
- The paper carefully and extensively positions itself in relation to existing work in this area. It is clearly written and easy to follow.
- Freely releasing the largest annotated dataset for NER/NED of tissue/cell types in and of itself is a large contribution.

**Additional Feedback:**

N/A.

**Correctness:**

- I don't think I agree with the claim on lines 22-24. Is it really LLMs driving advances in IE tasks like NER/NED/RE? I don't think so, as it seems smaller, fine-tuned models still dominate these tasks. Indeed many of the citations on this line are from pre-LLM era NLP. It might just be that we disagree on what an LLM is. I would feel more comfortable with this claim if "large language models" was swapped verbatim for "pre-trained language models (PLMs)."
- The paper often cites previous work using raw text strings of the author's names (e.g. line 154). I don't believe this is canonical. It would be better, I think, to properly cite things inline so readers can click the hyperlink and jump to the reference.

**Documentation:**

The dataset is extensively documented in the supplementary material, which includes a maintenance plan. The dataset is available online, freely, via GitHub.

**Limitations:**

I appreciated the inclusion of an explicit limitations section. I think it addresses any limitations I might have brought up.

**Opportunities For Improvement:**

- Is there a proper dev and test set? I don't see it mentioned in the paper or on GitHub. I think for future use this is important. Otherwise, researchers will slice up the data into train/dev/test in different ways, and evaluations might not be comparable.
- This is more of a nice to have, but have the authors added or considered adding FlaMBé to BigBIO? This would likely improve its usability and uptake in the research community. It would just involve writing a data-loading script and PR'ing it to the BigBIO repo.
- Why is NER performance reported per IOB tag? If anything, this is something I might expect to see in the appendix. In the main paper, why not report P/R/F1 for complete spans, which I think is easier to grok and closer to what we actually care about?

**Relation To Prior Work:**

Yes.

**Summary And Contributions:**

This paper presents a novel dataset, FlaMBé, which provided 55 full-text papers annotated for entity/relation extraction and disambiguation and >1000 abstracts labelled for entity extraction and disambiguation of tissue/cell types. Unlike previously labelled datasets for IE tasks in BioNLP, FlaMBé's focus is _procedural knowledge_ ("knowing how", as the authors put it).

---

> ### Author Response · Authors · 2023-08-23
>
> Thanks for the review and useful suggestions!
>
> > Is there a proper dev and test set? I don't see it mentioned in the paper or on GitHub. I think for future use this is important. Otherwise, researchers will slice up the data into train/dev/test in different ways, and evaluations might not be comparable.
>
> Thanks for this suggestion. We agree this will be important for future use and have added train/dev/test sets to the GitHub for iob files and will add the equivalent splits for the CoNLL files on Zenodo if we advance to acceptance. We have also updated the results in all tables in the paper to reflect these exact splits.
>
> > This is more of a nice to have, but have the authors added or considered adding FlaMBé to BigBIO? This would likely improve its usability and uptake in the research community. It would just involve writing a data-loading script and PR'ing it to the BigBIO repo.
>
> We think this is a great idea! While we have not yet added FlaMBe to BigBIO, this is certainly something that we will make a priority upon acceptance.
>
> > Why is NER performance reported per IOB tag? If anything, this is something I might expect to see in the appendix. In the main paper, why not report P/R/F1 for complete spans, which I think is easier to grok and closer to what we actually care about?
>
> We erred on the side of more detailed performance for our initial submission (also because for certain tasks, such as tool prediction, we observed a big difference between performance on B- vs I- tags), but agree with the reviewer that performance for complete spans are more interpretable. We have now changed our tables to report performance at the span-level.
>
> > I don't think I agree with the claim on lines 22-24. Is it really LLMs driving advances in IE tasks like NER/NED/RE? I don't think so, as it seems smaller, fine-tuned models still dominate these tasks. Indeed many of the citations on this line are from pre-LLM era NLP. It might just be that we disagree on what an LLM is. I would feel more comfortable with this claim if "large language models" was swapped verbatim for "pre-trained language models (PLMs)."
>
> We used the broad LLM terminology initially with the mindset that there are some categorizations that include the BERT-family of models under the LLM umbrella, but we agree that “pre-trained language models” is a more precise characterization and have edited the manuscript accordingly.
>
> > The paper often cites previous work using raw text strings of the author's names (e.g. line 154). I don't believe this is canonical. It would be better, I think, to properly cite things inline so readers can click the hyperlink and jump to the reference.
>
> We apologize for this oversight. We have edited all the inline citations to appropriately jump to the reference.

---

> > ### Comment · Reviewer_oXH1 · 2023-08-23
> > **Updated score**
> >
> > Awesome, thanks for addressing all my concerns! I don't have anything else to nitpick over, so I will go ahead and bump up my score. I just want to reiterate that I think adding this to BigBIO would be great for the community and might also increase FlaMBe's exposure!

---

### Official Review · Reviewer_oyUc · 2023-07-21
**Useful resource for understudied but important problem**

**Rating:** 7
**Confidence:** 3
**Clarity:** The paper is well written and easy to…

**Strengths:**

- The paper introduces FlaMBé a novel corpus with annotations for workflow extraction in the single cell literature. The extraction of procedural knowledge is an understudied but potentially very impactful task in (biomedical) information extraction. I commend the authors for providing a high-quality resource for this problem.
- Additionally, FlaMBé should be the largest available corpus for extracting (KB-linked) tissue/cell-type mentions from the biomedical literature.

**Additional Feedback:**

None

**Correctness:**

The data collection and experiments seem sound, albeit with some unconventional choices in experimental protocol.

**Documentation:**

The dataset is very well documented with an accompanying datasheet in the Appendix and a comprehensive Readme on github.

**Ethics:**

I don't have ethical concerns with this work.

**Limitations:**

The authors openly and adequately address the limitations of their work in a dedicated section.

**Opportunities For Improvement:**

- I could not locate the inter-annotator agreement for the workflow annotations. How did the authors perform quality control for these?
- The experiments are limited in scope. For instance, I think providing baseline results for workflow extraction would have been useful. However, I acknowledge that it would not be immediately clear what a good baseline method for this task would be. Maybe the authors could take inspiration from the biomedical event extraction literature (https://academic.oup.com/bioinformatics/article/36/19/4910/5858975)
- I find the choice to report separate metrics for B- and I-tags quite surprising. Conventionally, one would report span-level metrics, e.g. with seqeval (https://pypi.org/project/seqeval/).

**Relation To Prior Work:**

The paper clearly discusses the related work.

**Summary And Contributions:**

The paper ``Into the Single Cell Multiverse: an End-to-End Dataset for Procedural Knowledge Extraction in Biomedical Texts'' introduces FlaMBé a new dataset for extracting experimental workflows from single cell papers. FlaMBé contains annotations for tissue and named entities manually annotated and linked to a KB in 55 full-text papers and 1,195 abstracts. Additionally, it contains annotations for tools and their usage in the paper's workflow for the 55 full-text documents. The authors describe three possible uses of FlaMBé: (1) training Named Entity Recognition models, (2) predicting the context in which a tool was used and (3) visualizing employed workflows. They provide initial experiments for (1) and (2) and a visualization for (3).

---

> ### Author Response · Authors · 2023-08-23
>
> Thanks for your careful and thoughtful review!
>
> > I could not locate the inter-annotator agreement for the workflow annotations. How did the authors perform quality control for these?
>
> Unlike spans, the workflow annotations are more complicated to compare across annotators, and can be somewhat more subjective than spans with clearly defined boundaries. We believed that traditional metrics like Fleiss-Kappa would be unfairly harsh and difficult to adapt to adequately assess annotator agreement and thus chose to have all annotators annotate and review all workflow identified for each paper. This is also why we chose to annotate only a subset of full text papers. We opted to resolve annotation conflicts via round table discussion keeping only the consensus annotations, and resolving all other annotations only when a mutual agreement could be reached for their inclusion / exclusion from the final dataset. We have added text to the supplementary materials to clarify this further.
>
> > The experiments are limited in scope. For instance, I think providing baseline results for workflow extraction would have been useful. However, I acknowledge that it would not be immediately clear what a good baseline method for this task would be. Maybe the authors could take inspiration from the biomedical event extraction literature (https://academic.oup.com/bioinformatics/article/36/19/4910/5858975)
>
> We thank the reviewer for this suggestion: this is something that we agree would be valuable but, as pointed out, would require careful thought for an appropriate baseline. We hope that our dataset can be valuable to the community as is and we hope to tackle this topic as well as more thorough workflow predictions in a follow up study!
>
> > I find the choice to report separate metrics for B- and I-tags quite surprising. Conventionally, one would report span-level metrics, e.g. with seqeval (https://pypi.org/project/seqeval/).
>
> We erred on the side of more detailed performance for our initial submission (also because for certain tasks, such as tool prediction, we observed a big difference between performance on B- vs I- tags), but agree with the reviewer that performance for complete spans are more interpretable. We have now changed our tables to report performance at the span-level.

---

> > ### Comment · Reviewer_oyUc · 2023-08-29
> >
> > Thank you for your response. I think the limited scope of the experiments is still an open issue and thus I will leave my (positive) assessment unchanged.

---

### Official Review · Reviewer_oxgk · 2023-07-21
**Cell multiverse review**

**Rating:** 6
**Confidence:** 3
**Clarity:** Yes, the paper was clear and easy to …

**Strengths:**

Addresses an under-explored area in natural language processing (NLP): procedural knowledge extraction in scientific literature.
Dataset annotations are done by domain experts in computational biology, ensuring high quality and relevance.
Offers the largest NER and Named Entity Disambiguation (NED) dataset for tissues/cell types.
Provides annotations for software tools and computational methods, capturing 28 unique contexts and almost 400 workflow relations.
Demonstrates multiple use cases and potential downstream applications, which can benefit both the NLP and biomedical research communities.


**Additional Feedback:**

The paper presents a valuable dataset in an under-explored area of NLP and provides clear use cases and potential downstream applications. However, its focus on single-cell research limits its applicability to other biomedical domains, and some limitations, such as relatively fewer examples of complex annotation types, need to be addressed in future work. Also, a comparison with finetuned LLMs like LLAMA on this dataset would be an experiment worth running in the future.

**Correctness:**

The claims made in the submission seem correct, and the dataset appears to be constructed in a sound way. The evaluation methods and experiment design are appropriate, demonstrating the dataset's utility in various use cases and potential downstream applications.

**Documentation:**

Yes, the authors provide sufficient detail on data collection and organization, with a clear methodology for gathering and annotating the dataset. They also provide a link to a GitHub repository containing the dataset, making it available for exploration and download. Ethical and responsible use is discussed, with the authors encouraging thorough evaluation of predictions to avoid misleading downstream research.

**Ethics:**

No, the authors have acknowledged potential negative societal impacts and have provided suggestions to mitigate such risks. There are no apparent ethical concerns that warrant further discussion or review.

**Limitations:**

The authors acknowledge the dataset's limitations, particularly the relatively fewer examples of complex annotation types. They plan to address this issue through larger annotation efforts, potentially aided by computational predictions guiding manual curation efforts. The authors also discuss potential negative societal impacts, such as incorrect workflows possibly misleading downstream research. They encourage thorough evaluation of all predictions to mitigate this risk.


**Opportunities For Improvement:**

The dataset has relatively fewer examples of more complex annotation types, such as tool context and workflow.
The dataset mainly focuses on single-cell research, limiting its applicability to other biomedical domains.
The paper does not appear to have any evaluation comparisons to large language model finetuning which have shown to be extremely suitable for NER.

**Relation To Prior Work:**

The authors clearly discuss how FlaMBé differs from previous datasets and research efforts. FlaMBé is unique in capturing procedural knowledge in biomedical texts, focusing on computational analysis pipelines in single-cell research. It provides the largest NER and NED dataset for tissues/cell types and offers annotations for software tools, computational methods, tool contexts, and workflows. The paper also highlights the differences between FlaMBé and existing datasets in biomedical NLP and procedural knowledge extraction.

**Summary And Contributions:**

The paper presents FlaMBé (Flow annotations for Multiverse Biological entities), a dataset aimed at capturing procedural knowledge in biomedical texts, particularly focused on computational analysis pipelines in single-cell research. The dataset includes annotations for 1,195 paper abstracts and 55 full-text papers, covering nearly 700,000 tokens. It contains annotations for tissue/cell type, software tools, computational methods, tool contexts, and workflows. FlaMBé enables several use cases, such as Named Entity Recognition (NER), tool context prediction, and workflow visualization. The dataset's potential downstream applications include procedural knowledge extraction, method recommendation, and multiverse analyses for improved transparency and reproducibility in research.

---

> ### Author Response · Authors · 2023-08-23
>
> Thank you for your review!
>
> > The dataset has relatively fewer examples of more complex annotation types, such as tool context and workflow.
>
> This paper is, to our knowledge, the first of its kind for biomedical workflow extraction. Because this is a labor intensive, specialized task, we chose to assemble a small, but carefully, curated set of workflow annotations (with all annotators participating for each paper) with the hope that this dataset might inspire and aid in the creation of larger sets in this space.
>
> > The dataset mainly focuses on single-cell research, limiting its applicability to other biomedical domains.
>
> We appreciate the reviewer’s concern here, but respectfully think that instead, this specialization is a strength of our work. Single cell methods make up a sizable fraction of biomedical publications, covering over 615,000 papers on PubMed, which is growing rapidly with more than 140,000 papers published in this area since 2020. There is clearly a need for specialized mining and extracting methods in this space that will only continue to persist and grow. In addition, it is a space with particularly complex (and interesting) workflows, and recent large benchmarking efforts are beginning to highlight that navigating the “multiverse” of options may affect reproducibility and constitute another form of p-hacking, which is why we believe that this is also an incredibly valuable testbed for this type of procedural knowledge extraction task that could be eventually extended to other fields.
>
> > The paper does not appear to have any evaluation comparisons to large language model finetuning which have shown to be extremely suitable for NER.
>
> We agree that large language model finetuning is an important advance for NER. We opted for running BERT-based models because of their ease and historical use in the NER space to highlight the utility of our dataset. We hope to do a more thorough prediction focused paper in the future and use more fine-tuned LLMs for increased performance. Additionally, in response to the comments of reviewer oXH1, we have chosen to be more precise in our reference to BERT as one form of LLM and now refer to these models exclusively as pre-trained language models (PLMs).

---

### Official Review · Reviewer_bopq · 2023-07-21
**Review of "Into the Single Cell Multiverse: an End-to-End Dataset for Procedural Knowledge Extraction in Biomedical Texts"**

**Rating:** 9
**Confidence:** 4
**Correctness:** no issues here
**Clarity:** well written manuscript

**Strengths:**

- well-curated dataset, everything is done properly.
- sound annotation guidelines, datasheet and information about annotators provided in supplementary material
- clearly written manuscript
- sufficient positioning for related work
- three clear use cases of the dataset with experiments

**Additional Feedback:**

.

**Documentation:**

good documentation

**Ethics:**

no issues

**Opportunities For Improvement:**

- Figures 2 & 3: some description of how should the vertical bars be interpreted for the Sankey plot would be helpful
- Lines 247-248: why did the authors decide to use full texts as a training split and abstracts as a test split?
- Use case 3: can these visualisations be created automatically, or do they require manual annotations and plot configurations?
- I am not entirely sure if the comparison in Lines 254-262 is fair and makes sense. It looks like the model trained on Flambe has the advantage of using the same document structure.

**Relation To Prior Work:**

yes, related work is properly described

**Summary And Contributions:**

The submission introduces FlaMBé, a curated dataset for procedural knowledge extraction in biomedical texts, particularly in single cell research. It includes the largest named entity recognition (NER) and disambiguation (NED) datasets for tissue/cell types. This work creates valuable resource for biomedical NLP and presents three use cases with appropriate experiments. I believe that this will be a useful resource for the community. The dataset creation procedure is sound and clearly written.

---

> ### Author Response · Authors · 2023-08-23
>
> Thank you for your review and suggestions!
>
> > Figures 2 & 3: some description of how should the vertical bars be interpreted for the Sankey plot would be helpful
>
> We have added additional descriptions in Figures 2&3 to explain how to interpret the vertical bars in the revised manuscript.
>
> > Lines 247-248: why did the authors decide to use full texts as a training split and abstracts as a test split?
>
> Originally, our intention here was to highlight that predictive performance is generalizable between the full text and abstract compendia. In considering the reviewers’ comment, we realized that this design can be a bit confusing and unclear, and combined with Reviewer oyUc’s suggestion to release consistent train/val/test splits for reproducibility, we now provide train/val/test splits for both full text and abstracts.
>
> In the manuscript, we have updated the results to use a combination of full texts and abstracts for train/val/test (training partitions for each, validation partitions for each, etc). We hope that this improves clarity (and exploring predictive performance across full texts and abstracts (i.e., training on one / predicting on another) could be an interesting follow-up study)!
>
> > Use case 3: can these visualisations be created automatically, or do they require manual annotations and plot configurations?
>
> The sankey diagram visualizations can be created automatically (e.g., via d3). The figures presented in the manuscript were created “semi-automatically” - where small tweaks for visual clarity were added on top of an automatically generated sankey diagram. One of the downstream resources we may provide in the future is a means to explore annotated and predicted sankey diagrams via observable notebooks.
>
> > I am not entirely sure if the comparison in Lines 254-262 is fair and makes sense. It looks like the model trained on Flambe has the advantage of using the same document structure.
>
> We agree with the reviewer that one of the advantages of FlaMBe in this tool prediction evaluation is that the criteria used for annotations will naturally be more consistent when the model is trained on a subset of the FlaMBe tool annotations. However, we believe that this evaluation still is valuable and informative. The dataset we compare against, Softcite, has a similar goal (annotating mentions of software), and they also use full text research publications. Thus, both the models trained on FlaMBe and Softcite are using iob-formatted annotations on full text research publications, and in that sense, the broader structure is identical. Due to space constraints, in our original manuscript, we neglected to do a thorough discussion of the original Table 2 results, but we observed that the biggest performance difference between FlaMBe and Softcite was in the “I-TOOL” tags (i.e., tool names that spanned several tokens) and hypothesize that part of the advantage of FlaMBe’s tool annotations may be that it is solely focused on biomedical research publications (rather than both life sciences and economics as in Softcite), where there is a tendency to use very long names for software (and accompanying acronyms). We have added more details about both Softcite, our interpretation of the results, and the potential advantage of FlaMBe over Softcite due to similar annotation criteria (lines 255-269).

---

> > ### Comment · Reviewer_bopq · 2023-08-31
> >
> > Thank you for your response and the clarification. I have increased my score.

---

### Decision · Program_Chairs · 2023-09-22

**Decision:**

Accept (Spotlight)

**Comment:**

The reviewers all liked the paper. The authors' response clarified most points raised by the reviewers. In view of that, the authors are strongly invited to take the feedback on board for the final version.